# Long term intrinsic cycling in human life course antibody responses to influenza A(H3N2): an observational and modeling study

Bingyi Yang[1,2,3]*, Bernardo García-Carreras[1,2], Justin Lessler[4,5,6], Jonathan M Read[7], Huachen Zhu[8,9,10], C Jessica E Metcalf[11], James A Hay[12,13], Kin O Kwok[14,15,16], Ruiyun Shen[17], Chao Q Jiang[17], Yi Guan[8,9,10], Steven Riley[12]*, Derek A Cummings[1,2]*

[1]Department of Biology, University of Florida, Gainesville, United States; [2]Emerging Pathogens Institute, University of Florida, Gainesville, United States; [3]WHO Collaborating Centre for Infectious Disease Epidemiology and Control, School of Public Health, Li Ka Shing Faculty of Medicine, The University of Hong Kong, Hong Kong, China; [4]Department of Epidemiology, Johns Hopkins Bloomberg School of Public Health, Baltimore, United States; [5]Department of Epidemiology, UNC Gillings School of Global Public Health, Chapel Hill, United States; [6]UNC Carolina Population Center, Chapel Hill, United States; [7]Centre for Health Informatics Computing and Statistics, Lancaster University, Lancaster, United Kingdom; [8]Guangdong-Hong Kong Joint Laboratory of Emerging Infectious Diseases/MOE Joint Laboratory for International Collaboration in Virology and Emerging Infectious Diseases, Joint Institute of Virology (Shantou University/The University of Hong Kong), Shantou University, Shantou, China; [9]State Key Laboratory of Emerging Infectious Diseases / World Health Organization Influenza Reference Laboratory, School of Public Health, Li Ka Shing Faculty of Medicine, The University of Hong Kong, Hong Kong, China; [10]EKIH (Gewuzhikang) Pathogen Research Institute, Guangdong, China; [11]Department of Ecology and Evolutionary Biology, Princeton University, Princeton, United States; [12]MRC Centre for Global Infectious Disease Analysis, Department of Infectious Disease Epidemiology, School of Public Health, Imperial College London, London, United Kingdom; [13]Center for Communicable Disease Dynamics, Harvard TH Chan School of Public Health, Boston, United States; [14]The Jockey Club School of Public Health and Primary Care, Chinese University of Hong Kong, Hong Kong, China; [15]Stanley Ho Centre for Emerging Infectious Diseases, The Chinese University of Hong Kong, Hong Kong, China; [16]Shenzhen Research Institute of The Chinese University of Hong Kong, Guangdong, China; [17]Guangzhou No.12 Hospital, Guangzhou, Guangdong, China

**\*For correspondence:**
byyang@connect.hku.hk (BY);
s.riley@imperial.ac.uk (SR);
datc@ufl.edu (DAC)

## Abstract

**Background:** Over a life course, human adaptive immunity to antigenically mutable pathogens exhibits competitive and facilitative interactions. We hypothesize that such interactions may lead to cyclic dynamics in immune responses over a lifetime.

**Methods:** To investigate the cyclic behavior, we analyzed hemagglutination inhibition titers against 21 historical influenza A(H3N2) strains spanning 47 years from a cohort in Guangzhou, China, and applied Fourier spectrum analysis. To investigate possible biological mechanisms, we simulated

individual antibody profiles encompassing known feedbacks and interactions due to generally recognized immunological mechanisms.

**Results:** We demonstrated a long-term periodicity (about 24 years) in individual antibody responses. The reported cycles were robust to analytic and sampling approaches. Simulations suggested that individual-level cross-reaction between antigenically similar strains likely explains the reported cycle. We showed that the reported cycles are predictable at both individual and birth cohort level and that cohorts show a diversity of phases of these cycles. Phase of cycle was associated with the risk of seroconversion to circulating strains, after accounting for age and pre-existing titers of the circulating strains.

**Conclusions:** Our findings reveal the existence of long-term periodicities in individual antibody responses to A(H3N2). We hypothesize that these cycles are driven by preexisting antibody responses blunting responses to antigenically similar pathogens (by preventing infection and/or robust antibody responses upon infection), leading to reductions in antigen-specific responses over time until individual's increasing risk leads to an infection with an antigenically distant enough virus to generate a robust immune response. These findings could help disentangle cohort effects from individual-level exposure histories, improve our understanding of observed heterogeneous antibody responses to immunizations, and inform targeted vaccine strategy.

**Funding:** This study was supported by grants from the NIH R56AG048075 (DATC, JL), NIH R01AI114703 (DATC, BY), the Wellcome Trust 200861/Z/16/Z (SR), and 200187/Z/15/Z (SR). This work was also supported by research grants from Guangdong Government HZQB-KCZYZ-2021014 and 2019B121205009 (YG and HZ). DATC, JMR and SR acknowledge support from the National Institutes of Health Fogarty Institute (R01TW0008246). JMR acknowledges support from the Medical Research Council (MR/S004793/1) and the Engineering and Physical Sciences Research Council (EP/N014499/1). The funders had no role in the study design, data collection and analysis, decision to publish, or preparation of the manuscript.

## Editor's evaluation

This article follows the still unanswered concept of 'original antigenic sin' and shows the existence of a 24-year periodicity of the immune response against influenza H3N2. The valuable work suggests a long-term periodicity of individual antibody response to influenza A (H3N2) within a city.

## Introduction

Over a life course, a key feature of human adaptive immune responses is the ability to continually update and refine responses to new antigens. A key example is immune responses to influenza, a pathogen that is constantly experiencing genetic and antigenic change. Antibodies mounted against a specific influenza virus decay (in either absolute magnitude or antigenic relevance) after exposure until re-exposure or infection to an antigenically similar virus occurs, whereupon back-boosting of antibodies acquired from previous infections (e.g., activation of memory B cells) can occur, as well as updating antigen-specific antibodies to the newly encountered infection (e.g., activation of naïve B cells) (*Amanna et al., 2007*; *Edridge et al., 2020*; *Fonville et al., 2014*; *Kucharski et al., 2018*). As antibodies are considered a correlate of protection from infection (*Cowling et al., 2019*; *Dunning, 2006*; *Krammer, 2019*; *Truelove et al., 2016*), studies often measure antibodies against the circulating strain to estimate the risk of infection. However, interactions between antibodies that were acquired from recent and long-ago infections can mean that characterization of antibodies to only currently circulating strains of pathogens may only partially capture antibody protection and risk of infection (*Cowling et al., 2019*; *Ng et al., 2019*; *Yang et al., 2020*).

Original antigenic sin (OAS) is a widely accepted concept describing the hierarchical and persistent memory of antibodies from the primary exposure to a pathogen in childhood. Recent studies suggested that non-neutralizing antibodies acquired from previous exposures can be boosted and may blunt the immune responses to new influenza infections (e.g., immunodominance) (*Andrews et al., 2015*; *Auladell et al., 2022*; *Gouma et al., 2020*; *Krammer, 2019*). Antibody-mediated immune response to multiple infections generated through repeated exposures to antigenically variable pathogens results in not only the facilitative interactions (e.g., back-boosting and cross-protection; *Krammer, 2019*), but

also competitive interactions (e.g., immune imprinting; *Gostic et al., 2016*; *Reynolds et al., 2022*; *Vieira et al., 2021*), and antigenic seniority (*Lessler et al., 2012*). Immune functions targeting antigenically specific pathogens may rise or lower in prevalence over a person's lifetime, in response to both a new infection and these competitive and facilitative interactions. Such interactions provide positive and negative feedbacks that have routinely been found to drive cycles in other systems (e.g., predator–prey, host–parasite) (*Post and Palkovacs, 2009*; *Yoshida et al., 2003*). Therefore, we might expect feedback mechanisms to introduce intrinsic temporal cycles in an individual's immune responses to antigenically variable pathogens over a lifetime, yet these cycles have not often been investigated.

Here, we examine seasonal influenza as a case study. Three subtypes of influenza (A(H3N2), A(H1N1), and B) cause an estimated 291,000–645,000 deaths globally every year (*Iuliano et al., 2018*). Although viruses of the same subtype share similar surface proteins, continuous genetic mutation leads to antigenic variation, resulting in escape from immune recognition by antibodies generated by previous infections. However, escape is not complete. Cross-reactive immunity across strains exists for viruses isolated at different times (*Bedford et al., 2014*; *Fonville et al., 2014*; *Krammer, 2019*; *Smith et al., 2004*). While high levels of antibody have been found to be protective from infection, they have also been found to be associated with reduced antibody responses to new infections and influenza vaccination (*Auladell et al., 2022*). New infections were found to boost antibodies against previously encountered viruses as much if not more than that of the infecting virus (*Auladell et al., 2022*). Therefore, we hypothesized that the combination of antigen-specific and nonspecific responses may give rise to cycles in antibody responses over an individual's life span. We tested the hypothesis that human adaptive immune responses exhibit nonlinear interactions with evolving viruses, creating intrinsic cycles in antibody responses.

To test the hypothesis, we characterized the periodic behavior of 777 paired antibody profiles, measured in 2010 (baseline) and 2014 (follow-up), measuring antibody responses (i.e., hemagglutination inhibition [HI] titers) to 21 A(H3N2) strains circulating over 47 years (*Figure 1A*, *Figure 1—figure supplements 1–3*; *Jiang et al., 2017*; *Yang et al., 2020*). Only 0.6% (n = 5) of participants self-reported influenza vaccinations between the two visits; therefore, the observed changes in HI titers between the two visits were likely due to natural exposures. We used Fourier analysis to examine the periodicity of individual antibody responses, after accounting for shared variations arising from virus-specific population-level circulation and/or laboratory measurement. We assessed the robustness of the observed cycles to multiple analytic and sampling methods. We then used a previously published mechanistic model that characterizes individual antibody responses to a set of antigenically similar strains to test the sensitivity of these cycles to multiple generally recognized biological mechanisms (*Kucharski et al., 2018*). Finally, we determined whether the cyclic pattern in individual antibody responses is predictable and whether it could improve the prediction of the risk of seroconversion to circulating strains of influenza A(H3N2) over existing models.

## Results

### Identifying long-term cycles in individual antibody responses to influenza A(H3N2)

Antibody titers against a set of strains isolated over 47 years, when ordered by the time of isolation of the tested strains, form a time series that describes the immune history of an individual and cover a range of antigenic distances (*Figure 1—figure supplements 3 and 4*; *Yang et al., 2020*). To describe variations in these time series attributable to virus-specific and/or individual-level host characteristics, we fitted a generalized additive model (GAM) of log-titers with strain-specific intercepts and nonlinear effects of age at serum collection (i.e., biological age) and age at the year when strains were isolated (i.e., birth cohort effect) (*Kucharski et al., 2018*). Strain-specific intercepts (*Figure 1B*, *Figure 1—figure supplement 1B*) were estimated to adjust for the average population antibody responses due to A(H3N2) circulation and/or virus-specific differences in laboratory assay measurements. Residuals were then estimated to represent individual-level departures from population averages (*Figure 1C*, *Figure 1—figure supplement 5*) and were interpolated to annual resolution with spline function (see details in 'Methods).

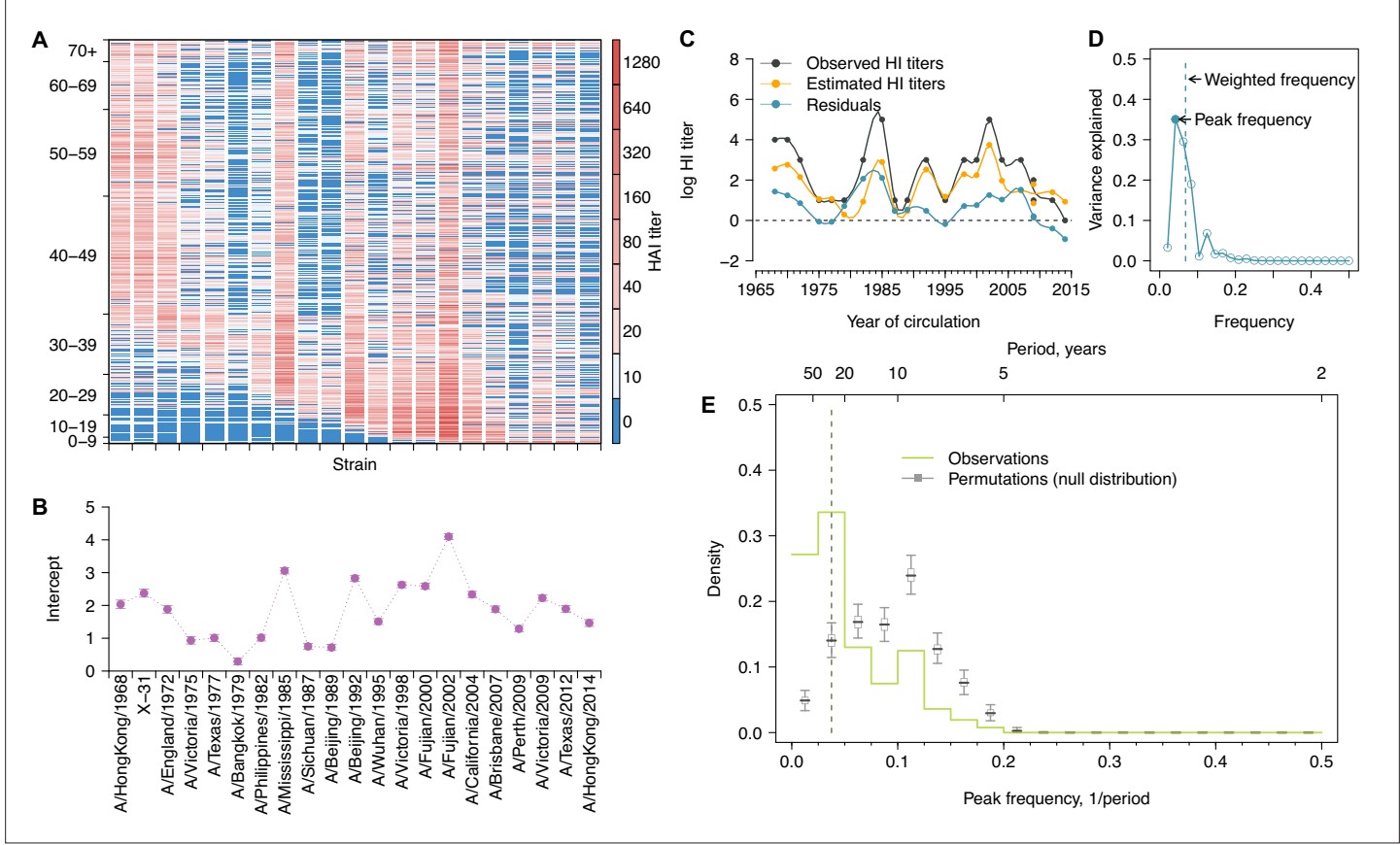

**Figure 1.** Long-term cycles in individual antibody responses to influenza A(H3N2) at baseline. (**A**) Hemagglutination inhibition (HI) titers against A(H3N2) strains at baseline. Each row shows an antibody profile for a participant. Participants are sorted by age (y-axis). Strains (x-axis) are sorted by the year of isolation, which are listed in the x-axis of (**B**). (**B**) Strain-specific intercepts. A generalized additive model (GAM) was fitted to log HI titers (shown in **A**) on age at sampling (spline), age at isolation (spline), and strains (categorical) (also used for **C**). With the model, we extracted strain-specific intercepts (representing population level activity; shown in **B**) and calculated the residuals between predicted and observed log HI titers for each individual (individual-level antibody responses; shown in **C** and used for **D, E**; details in *Figure 1—figure supplement 5A*). (**C**) Illustration of estimating individual time series of residuals. Estimates were derived from the GAM model in (**B**). Residuals were calculated as the difference between observed and estimated HI titers (i.e., black minus orange; shown as the blue line). (**D**) Illustration of a Fourier spectrum. Peak (i.e., the frequency explaining the largest variance) and weighted frequency of a Fourier spectrum of the interpolated time series of residuals shown in (**C**). (**E**) Distribution of peak frequencies of individual residuals. We performed Fourier spectral analysis (shown in **D**) on the time series of residuals of each person and extracted the peak frequency. The light green shows the distribution of peak frequencies across participants, with the dashed vertical line indicating the peak frequencies that had the highest proportions among individuals. Median (thick gray ticks), interquartile (gray boxes), and 95% intervals (thin gray ticks) of distributions from 1000 permutations.

The online version of this article includes the following source data and figure supplement(s) for figure 1:

**Source data 1.** Variance (%) explained by low frequencies and peak frequencies for Fourier spectra of individual residuals.

**Figure supplement 1.** Long-term cycles in individual antibody responses to influenza A(H3N2) at follow-up.

**Figure supplement 2.** Representative individual profiles of hemagglutination inhibition (HI) titers.

**Figure supplement 3.** Antigenic map and paired antigenic distances of influenza A(H3N2) strains.

**Figure supplement 4.** Conceptual plot for individual life-course immune responses to influenza.

**Figure supplement 5.** Representative individual profiles of residuals of hemagglutination inhibition (HI) titers.

**Figure supplement 6.** Illustration of estimation of individual time series of residuals and Fourier analysis of observed and permutation of time series.

**Figure supplement 7.** Validation using values generated from random distributions with no periodicity.

**Figure supplement 8.** Validation using values generated from periodic curves.

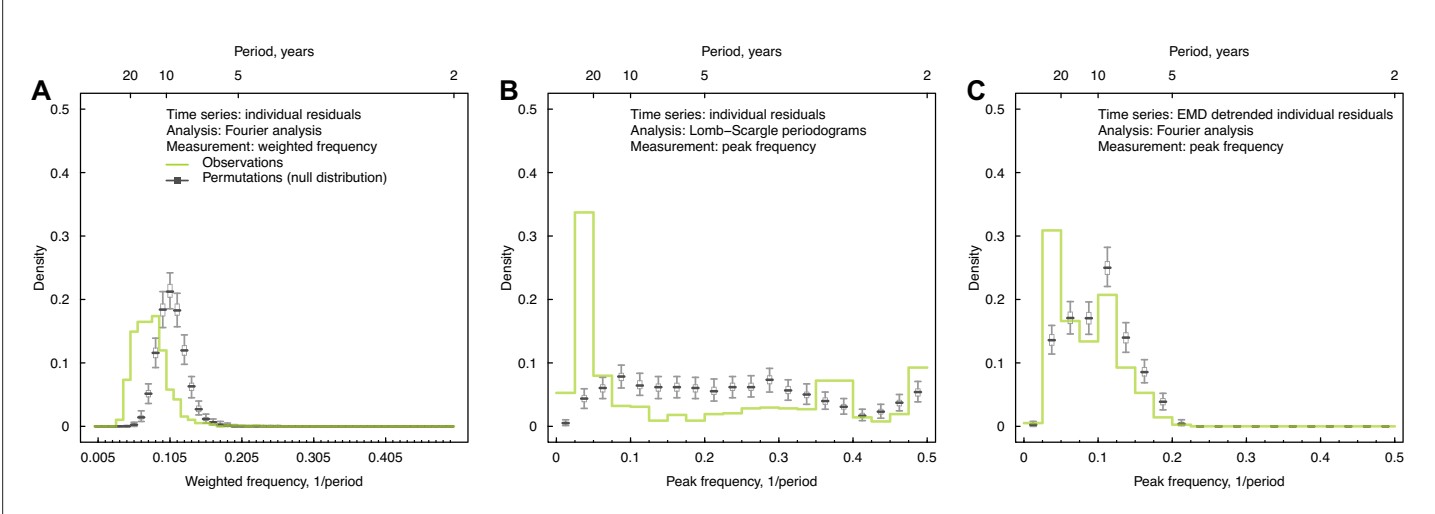

**Figure 2.** Impacts of irregularly sampled data, interpolations, and long-term trends on cycles identified in individual antibody responses at baseline. (**A**) Distribution of weighted frequencies of individual Fourier spectra at baseline. We performed Fourier spectral analysis on the interpolated time series of residuals for each person and calculated the average frequency weighted by the variance explained ('weighted frequency'; see **Figure 1D**). (**B**) Distribution of peak frequencies of individual Lomb–Scargle periodograms. We performed Lomb–Scargle periodograms on the time series of residuals for each person and extracted the frequency that explained the most variance ('peak frequency'). (**C**) Distribution of peak frequencies of individual Fourier spectra of detrended residuals at baseline. We performed Fourier analysis on time series that removed the nonlinear trend identified using empirical mode decomposition (EMD) analysis.

The online version of this article includes the following figure supplement(s) for figure 2:

**Figure supplement 1.** Comparison of time series of residuals and their Fourier spectra before and after empirical mode decomposition (EMD).

**Figure supplement 2.** Distribution of peak frequencies across individuals when dropping titers of one tested strain for serums collected at baseline.

**Figure supplement 3.** Distribution of peak frequencies across individuals when dropping titers of one tested strain for serums collected at follow-up.

**Figure supplement 4.** Distribution of peak frequencies across individuals born before 1968.

**Figure supplement 5.** Cycles in immune responses to influenza in the Vietnam data.

We investigated whether cyclic behavior was present in antibody responses by performing Fourier analysis on each individual's time series of residuals (**Figure 1—figure supplement 5**). The periodicity for each participant was determined by the frequency ('peak frequency' hereafter) that explained the most variance in the Fourier spectrum (**Figure 1D**). To test the significance of these peak frequencies, we compared the distribution of peak frequency across participants with those distributions (i.e., null expectation) from 1000 permutations, in which observations for each time series were shuffled (**Figure 1—figure supplement 6C and D**). This null expectation represents the peak frequency distribution of random nonperiodic time series and reflects the underlying structure that is introduced by our sampling and interpolation approaches (**Figure 1—figure supplements 7 and 8**).

We found that 33.6% (95% CI, 30.3–37.0%) of participants had a peak frequency corresponding to a long-term periodicity (i.e., 20–40 years, translated from frequencies of 0.025–0.050; see 'Methods') at baseline, which was significantly higher than null expectation (**Figure 1E**), suggesting that the observed cyclic patterns were not purely due to chance. This peak frequency range (corresponded to an ~24-year periodicity) accounted for a median 22.1% of the variance (interquartile range [IQR], 11.1–35.4%) of individual-level residuals (**Figure 1—source data 1**). We observed similar periodicity at follow-up, indicating that such pattern was unlikely to be affected by recent exposures (**Figure 1—figure supplement 1C**).

We conducted multiple sensitivity analyses and validations to test the robustness of the observed cycles in individual-level antibody responses to analytic methods and our sampling methods. Across these analyses, including methods that accounted for variation in each individual's spectra (**Figure 2A**), irregularity in isolation intervals of tested strains (**Figure 2B**), and secular trends in our time series (**Figure 2C**, **Figure 2—figure supplement 1**), we found consistent evidence for long-term periodicity in antibody responses. Results were robust to leaving specific strains out of the analysis

(*Figure 2—figure supplements 2 and 3*) and method of interpolation (*Figure 1—figure supplements 7 and 8*). A full description of sensitivity analyses including validation in subsets of our data is provided in 'Methods' (*Figure 1—figure supplements 7 and 8*, *Figure 2—figure supplements 2–4*).

Additionally, we analyzed an independent out-of-sample data set from Vietnam (HI titers of 57 strains for 69 participants measured annually, 2007–2012; *Horby et al., 2012*; *Kucharski et al., 2018*). Due to the lack of data on age, we compared long-term periodicity in HI titers and found a similar long-term periodicity in both studies (*Figure 2—figure supplement 5*), suggesting that similar cycling is likely present in other settings, even with population-level variations.

## Cycles in individual antibody responses likely associated with homotypic cross-immunity

To investigate possible biological mechanisms, we simulated individual antibody profiles encompassing known feedbacks and interactions due to generally recognized immunological mechanisms (*Figure 3—source data 1 and 2*). We primarily applied a model by *Kucharski et al., 2015* that describes the snapshot of individual antibody dynamics, resulting from varied individual infection histories, narrow (i.e., against recent strains) and broad (i.e., against distant strains) range of cross-reactions of antigenically similar strains and antibody waning (*Figure 3—figure supplement 1*; *Equation 9*). We extended the model to allow for the influence of individual-level preexisting antibodies and population-level viral activity on individual infection hazard (*Figure 3I*, *Figure 3—figure supplement 1*; *Equations 10 and 11*). Infection events are simulated annually and individually according to individual infection hazard, which is then used to inform the updated antibody profiles using Kucharski's model (*Figure 3—figure supplement 1*). As viral circulating pattern at population level is not the focus of this study and its potential drivers (e.g., arising from homotypic and/or heterotypic cross-immunity) are inconclusive, we therefore assumed two scenarios (*Figure 3I*) to examine the impact of predictable (i.e., cyclic) or nonpredictable (i.e., random) annual attack rates on the observed individual antibody responses.

We simulated individual infection histories since 1968 and sampled these simulated histories with the same time resolution as tested strains measured in 2014. We applied Fourier analysis on the resulting individual time series (see 'Methods'). We tested several potential biological mechanisms that can shape individual antibody profiles through influencing individual infection hazard (i.e., individual preexisting titer to the circulating strain and population-level circulation) and antibody responses after exposures (i.e., broad and narrow cross-reactions) (*Figure 3I*, *Figure 3—figure supplement 1*). The breadth of such cross-reactions was implicitly assumed to be determined by the antigenic evolution rate in our simulations, which is 0.778-unit changes in the antigenic space per year according to prior estimates (*Fonville et al., 2014*; *Kucharski et al., 2018*).

We assessed the periodic pattern of the simulation from two perspectives. First, we compared whether the peak frequency distribution from the simulation was significantly different from the null distribution to determine whether the simulated antibody profiles were periodic (*Figure 3I*). Next, we compared whether the simulated antibody responses had a higher proportion of peak frequency of 0.025–0.050 compared with the null distribution, to determine whether the simulations could recover the long-term periodicity that was identified in the empirical data.

Multiple models showed qualitatively similar periodic behavior to data that is different from null distribution and had a significantly higher proportion of simulated individual responses with long-term periodicity (*Figure 3*). A key model component that exhibited long-term periodicity was cross-reactivity between antigenically similar viruses, especially broad-range (i.e., against distantly related strains) cross-reactions (*Figure 3D–H*). When the component of broad-range cross-reactions was absent in the model, population-level circulation alone was not able to recover the long-term periodicity in individual antibody responses (*Figure 3B*). However, when cross-reaction in antibody responses was included in the model, a less predictable population-level activity (i.e., random compared to cyclic variation, *Figure 3G and H*) appeared to introduce more uncertainties in the observed cycles in individual antibody responses.

## Predicting seroconversion to recent strains using cycles in individual antibody responses

These results suggested that, after accounting for the impact of population-level A(H3N2) circulation, cross-reactivity from previously infected strains likely explained the reported cyclic patterns in an

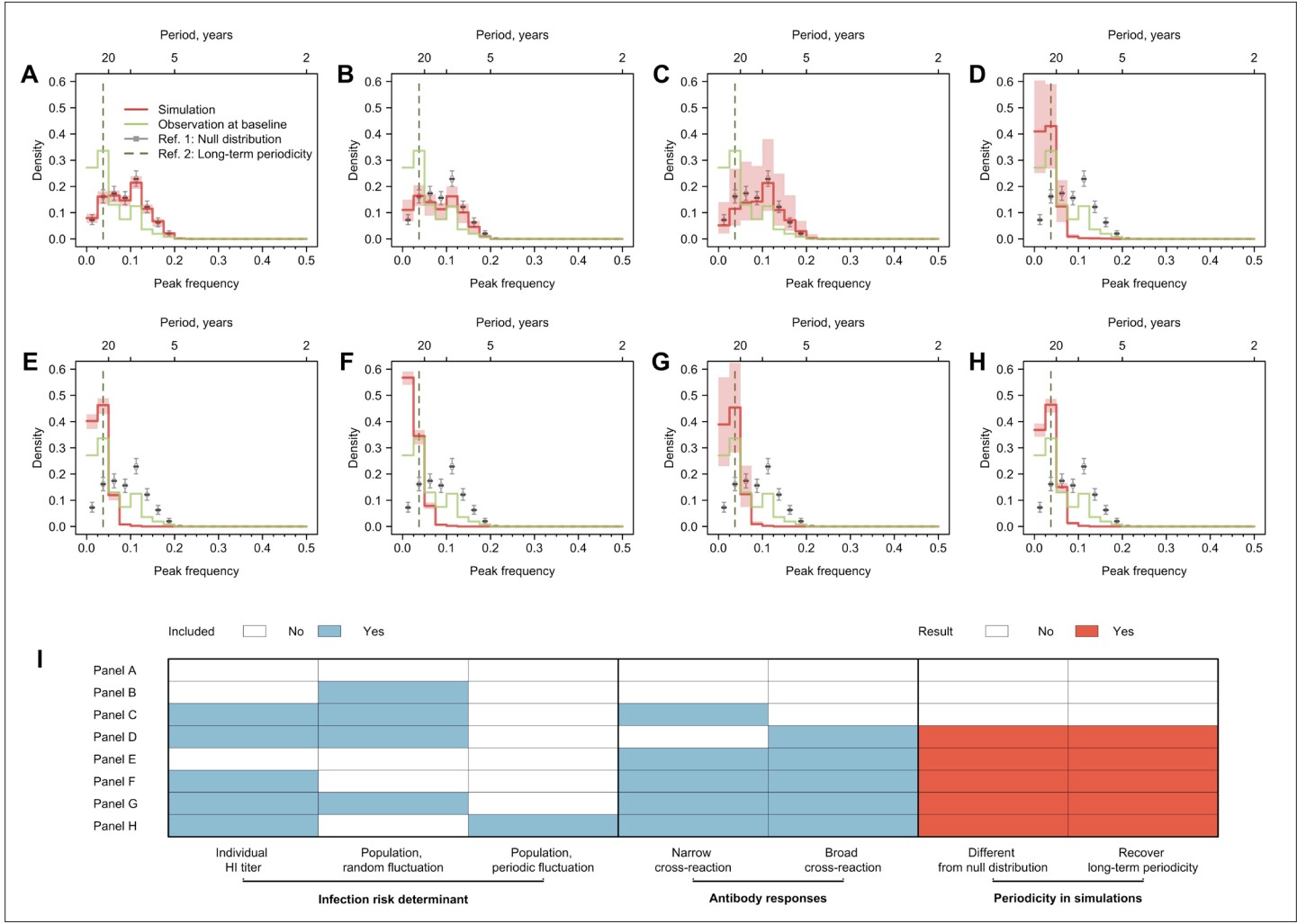

**Figure 3.** Cycles in simulated antibody responses from the model accounted for different mechanisms. Colored lines are the distribution of peak frequencies detected in the simulated antibody profiles across individuals. Gray lines are the distributions of peak frequencies of the 1000 permutations of the simulated antibody profiles. For each scenario, we simulated the life course of infections and immune responses for 777 individuals of the same age as the participants in our study and extracted the antibody profile in 2014 for the year's corresponding to when our 20 strains were isolated. (**A**) No biological mechanisms were modeled, and the individual risk of infection each year was purely random with a fixed probability of 0.2. (**B**) Narrow (i.e., against antigenically similar strains) and broad (i.e., against distant strains) cross-reactions of antibodies were modeled, which would however not affect individual risk of infection every year (i.e., the risk of infection each year was purely random with a fixed probability of 0.2). (**C**) Individual risk of infections was modeled as the randomly varied population-level H3N2 activity every year (i.e., not affected by individual antibody responses), no cross-reactions of antibodies were modeled. (**D–F**) Narrow and broad cross-reactions of antibodies were modeled, with greater cross-reactions conferring higher level of protection. Population-level H3N2 activity were modeled as constant (**D**), randomly (**E**), and periodically (**F**) varied, respectively. (**G**) Broad cross-reactions of antibodies were modeled, with greater cross-reactions conferring higher level of protection. Random variations in population-level H3N2 activity were modeled. (**H**) Narrow cross-reactions of antibodies were modeled, with greater cross-reactions conferring higher level of protection. Random variations in population-level H3N2 activity were modeled. (**I**) Biological mechanisms included in models that generated results in (**A–H**).

The online version of this article includes the following source data and figure supplement(s) for figure 3:

**Source data 1.** Parameters used in the simulations.

**Source data 2.** Mechanisms examined in the simulations.

**Figure supplement 1.** Conceptual plot of modeling immune responses.

**Figure supplement 2.** Impact of antigenic evolution speed on the reported cycles in individual antibody responses.

individual's antibody responses. As such, we hypothesized (1) that the position of individuals in their antibody response cycles could be predicted years in advance if the periodic behavior was stable over 3–4 years and (2) that the position of individuals in their antibody response cycles are associated with responses to future strains. We measured the position in antibody response cycles using phase angles (*Figure 4A*).

To test the first hypothesis, we predicted the phase of individual antibody response residuals to strains circulating in 2012 (midpoint of baseline and follow-up; *Figure 4B*) by fitting a log-linear regression to the residuals of HI titers measured at baseline (2010) against 14 historical strains (i.e., isolated between 1968 and 2002) on harmonic terms that represent the long-term periodicity (assuming as 24 years). We found high consistency between predicted and observed phases in 2012 across participants (*Figure 4C*). For example, a consistency of 73% (95% CI, 65–79%) among individuals whose antibody responses were predicted to be in phase I (i.e., the first quarter of a cycle).

To test the second hypothesis, we fitted a logistic regression of seroconversion (i.e., fourfold rise in HI titers to A/Texas/2012 or A/HongKong/2014) between baseline and follow-up on the above-mentioned predicted phase in 2012 (using strains isolated between 1968 and 2002), and adjusted for biological age at baseline and the average preexisting log-titer of the two strains. We found that individuals who were predicted to be in phase IV (i.e., the last quarter of a cycle) were 14% (95% CI, 4–26%) more likely to experience seroconversion to the two recent strains compared to those in phase I (*Figure 4D*).

## Disentangling cohort effects using cycles in individual antibody responses

As a result of resonance, we expected intrinsic cycles in individual antibody responses to be correlated across birth cohorts (*Figure 4E*). There were indications of this in the correlation of phase across cohorts. For example, we observed a higher proportion of participants who were in phase IV for the 2012 strain at baseline, when comparing the birth cohorts of 1986–90 (55%, 95% CI, 39–70%) with 1961–65 (17%, 95% CI, 11–25%; chi-squared test, p<0.001). Moreover, we found that such cohort-specific differences in phase composition (Pearson correlation = 0.67, p=0.01; *Figure 4F*) and the resulting proportions of seroconversion (Pearson correlation = 0.63, p=0.02; *Figure 4G*) correlated with the predicted cohort-specific composition of phase IV in 2012. Such correlations disappeared when assuming a 35- or 6-year periodicity (i.e., periodicities that were not supported by observations in *Figure 1* and *Figure 4—figure supplement 1*). Of note, we found that a diversity of phases was exhibited by members of the same age cohorts, suggesting that individual's cycles could depart from other members of their birth cohort.

## Discussion

We demonstrate that human antibody responses to influenza A(H3N2) display long-term periodicity, which are biologically consistent with nonlinear human adaptive immune responses (i.e., cross-reactions) to evolving viruses. Our observations are validated by different analytic methods and validation in a separate study population tested by a different antibody assay. Our findings were robust to our sampling and interpolation methods. We further demonstrate that, at both individual and birth cohort levels, the phases of the antibody responses to the currently circulating strains are predictable and associated with seroconversion to these strains independent from the preexisting titers and age. Such findings could improve our forecasting of the individual and birth cohort-level risks of infections and our understanding of heterogenous immune responses and vaccine effectiveness against influenza viruses.

We were able to qualitatively recover the observed long-term periodicity only when including cross-reactions between antigenically similar strains in the simulations. Particularly, our simulation results suggested that model including repeated exposures or population-level A(H3N2) activity alone did not recover the long-term periodicity (*Figure 3*). Such findings fit in previous observations that strain-transcending antibody responses to past infections accumulate and build up contemporary antibody profiles (*Fonville et al., 2014*; *Yang et al., 2020*). Of note, the long-term periodicity is a retrospective characterization of individual antibody profiles that arose from multiple exposures

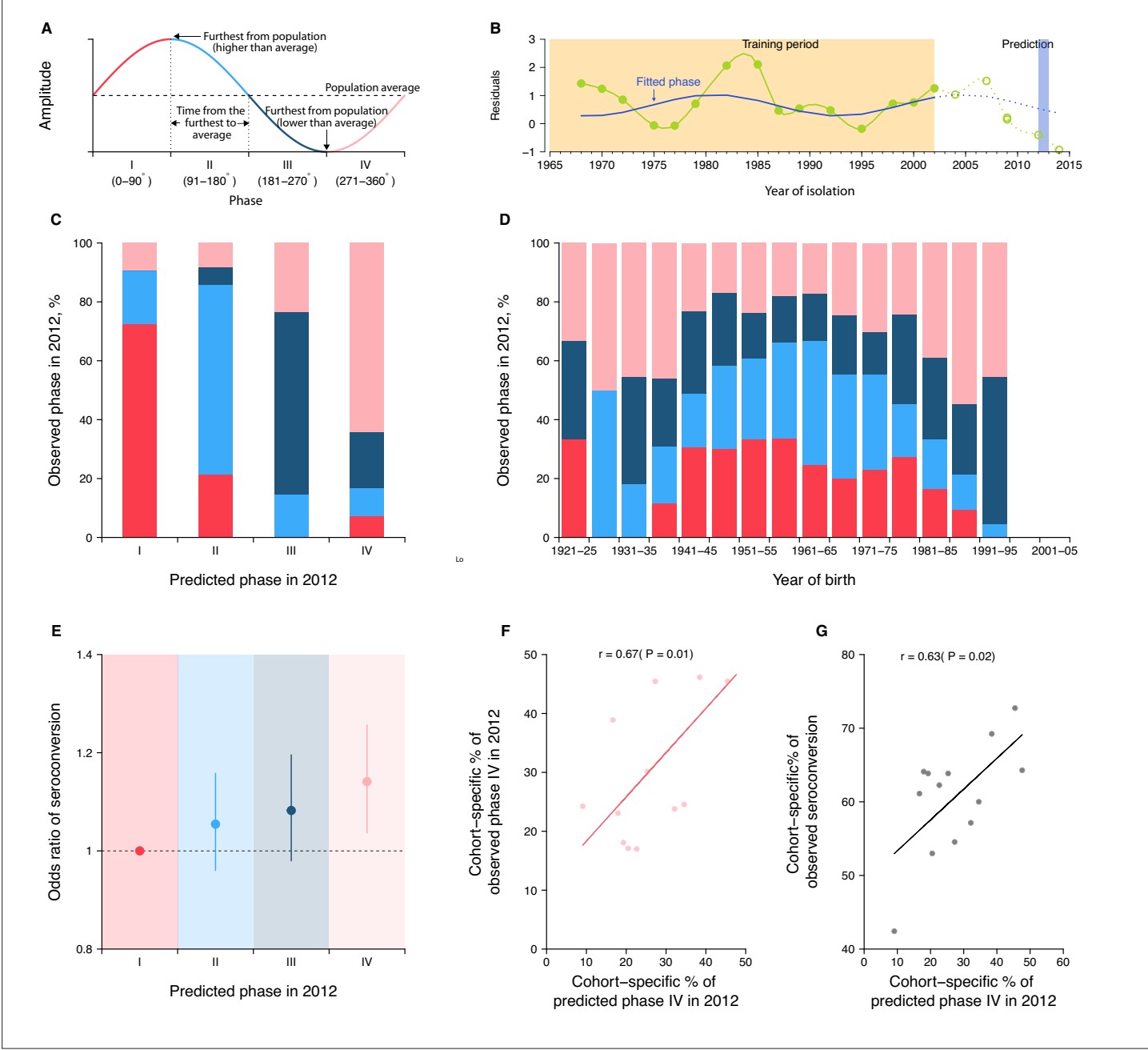

**Figure 4.** Predicting seroconversion to the recently circulated strains using phases of individual antibody responses. (**A**) Concept plot for phases. Four phases were classified based on the phase angles between 0 and 360°. (**B**) Illustration of predicting phase in 2012 using individual residuals from baseline hemagglutination inhibition (HI) titer that were measured against 14 historical strains (i.e., isolation year up to 2002). Green dots and solid green lines indicate the residuals against historical strains that were used to fit the periodic function (shown in solid blue lines). With the fitted periodic function, we predicted phase angles in 2012 based on the predicted residuals for individual's titers against strains that were circulating after the training period (shown in dotted blue lines). For reference, we also showed the observed baseline residuals for individual's titers against strains that were circulating after the training period (indicated as green circles and dotted green lines). (**C**) Observed and predicted phase in 2012 across participants. Colors represent participants' observed phase in 2012, with I, II, III, and IV represented by red, light blue, dark blue, and pink, respectively (same for **D** and **E**). (**D**) Observed cohort-specific distribution of phase in 2012. (**E**) Adjusted risk of seroconversion to recent strains (i.e., A/Texas 2012 or A/HongKong/2014) between baseline and follow-up visits for different phases. We estimated associations between phases in 2012 that were estimated from individual antibody profile residuals and seroconversion to any of the two recent strains and adjusted for age at sampling and the average preexisting titer of the two strains. (**F**) Observed and predicted cohort-specific proportion of phase IV in 2012. (**G**) Predicted proportion of phase IV in 2012 and the observed proportion of seroconversion to recent strains between baseline and follow-up for each cohort.

*Figure 4 continued on next page*

*Figure 4 continued*

The online version of this article includes the following figure supplement(s) for figure 4:

**Figure supplement 1.** Impact of periodicity on predictions of cohort-specific phases and seroconversions to circulating strains.

and cross-protection, which should not be directly interpreted as the duration of onward protection conferred by existing antibodies.

Within-subtype cross-reactions may drive individual-level long-term periodicity in antibody responses through temporal (but waning) cross-protection (i.e., positive feedback) and blunting generation of specific antibodies (i.e., negative feedback) against the circulating strains. A recent cohort study found that homotypic cross-protection against PCR-confirmed infections for up to five seasons after infections supported cross-protection that eventually wanes (*Wraith et al., 2022*). In addition, the ~24-year periodicity implicitly suggested that antibodies gained from last immunizing events may interfere with the antibodies against the circulating strains for a maximum of 18 years (i.e., phases I–III in *Figure 4B*), before the antigenicity between the last immune strain and the circulating strain was too different to cause high-level cross-reactions in binding antibodies. This is in line with our previous findings that people's sera showed very little to no cross-reaction with strains that were isolated 20 years prior to their births (*Lessler et al., 2012*; *Yang et al., 2020*).

Our findings suggest that long-term periodicity in HI titer may be driven by broad cross-reactions between strains that accumulate as people are exposed to multiple viruses over their lives. The breadth of such cross-reactions was determined by previously reported antigenic evolution (*Fonville et al., 2014*; *Kucharski et al., 2018*). In simulations, we found that antigenic evolution rates significantly change the periodicity in individual antibody responses (*Figure 3—figure supplement 2*). Slower antigenic evolution rates shift cycles in individual antibody responses to longer periodicity, with the extreme that people could acquire lifelong immunity against antigenically stable viruses (e.g., measles) (*Amanna et al., 2007*). Though in our simulations faster antigenic evolution led to shorter cycles, high rates of antigenic evolution could diminish the periodicity in antibody responses through frequent reactions to re-exposures.

We found associations between the phase of antibody response cycles and the risk of seroconversion to circulating strains after accounting for the homologous preexisting HI titers. Due to the low influenza vaccine coverage in our participants and in China in general, the observed seroconversions likely reflected antibody responses after natural exposures during the study period. Previous studies have reported differential risk in individuals with the same homologous HI titer, proposing that unknown individual exposure histories and cohort effects are possible explanations (*Turbelin et al., 2013*; *Yang et al., 2018*). Our findings suggest that cyclic patterns in an individual's antibody responses, which may be predictable at both individual and cohort levels, may contribute to this heterogeneity. In addition, our findings suggested that measuring seroconversion against a circulating strain could reveal a limited amount about protective immunity. Measuring responses to both circulating and previously circulating viruses (as well as calculating phase) could improve characterization of people's risk (*Quandelacy et al., 2021*).

We demonstrate that resonance of cycles in individual-level antibody responses could form variations in phase distribution of antibody responses across birth cohorts, which is consistent with previous findings that the fraction of A(H3N2) associated cases across different birth cohorts was found to change year to year (*Turbelin et al., 2013*; *Yang et al., 2018*). Our results also showed that the phase distribution of antibody responses across birth cohorts may be predictable and could be further used to predict the cohort-specific seroconversion against the circulating strains. This is potentially useful to determine and vaccinate the high-risk groups based on the exposure histories that could be shared by birth cohort. We did not attempt to explore the impact of antigenic evolution speed and the associated dynamics in antibody responses on shaping age-specific patterns of cases, while we speculate that antigens with faster antigenic evolutions may attack different age groups at relatively similar risks, while antigens with slower antigenic evolutions tended to attack the children (e.g., A(H3N2) vs. B/Victoria in *Turbelin et al., 2013*; *Yang et al., 2018*).

In this study, we did not explore the interactions between individual-level antibody responses with population-level A(H3N2) activity (e.g., epidemic sizes). We minimized the impacts from population level by performing the Fourier analysis with individual departures from population average and validating the results with data from the Vietnam cohort. Simulation results further suggested that

the population-level virus activity alone was not able to recover the observed periodicity, though epidemics with less regularity seemed to increase the variability in individual-level periodicity in the presence of broad cross-reactions (*Figure 3G and H*).

We recognize that HA-binding antibodies only mediate about half of the protection against influenza infections, while other forms of immunity, including neutralizing antibodies, non-HA head-specific inhibitory antibodies, and cellular immunity, would provide independent protections against infection and the severity of the diseases (*Cowling et al., 2019*; *Meade et al., 2020*; *Ng et al., 2019*; *Stadlbauer et al., 2019*). The breadths and oscillation patterns may differ across different forms of immunity.

Our work has several limitations. First, between-subtype interactions have not been incorporated into our framework. It is arguable that whether infections with A(H1N1) could confer years-long cross-protections against A(H3N2) in humans at individual level (*Sonoguchi et al., 1985*), while prior findings tended to support that between-subtype interaction could alter the transient cycles (i.e., within seasons) (*Goldstein et al., 2011*; *Meade et al., 2020*; *Ranjeva et al., 2019*). Nevertheless, our simulation results suggested that only including population-level circulation – regardless of its underlying drivers – could not recover the observed long-term periodicities in individual antibody responses. Second, our simulation results, while robust across parameter settings, may depend on the simplifying assumptions on immunological mechanisms we made (e.g., individual immunity-dependent protection) and therefore only qualitatively recovered the observed pattern. Finally, the exact value for the long-term periodicity was determined by a series of fixed frequencies that were examined in the Fourier analysis, which depend on the number and span of the tested strains. Therefore, more accurate values for the periodicity could be estimated if the tested strains were sampled more densely.

## Methods

### Ethical approval

The following institutional review boards approved the study protocols: Johns Hopkins Bloomberg School of Public Health (IRB 1716), University of Florida (IRB201601953), University of Liverpool, University of Hong Kong (UW 09-020), and Guangzhou No. 12 Hospital ('Research on human influenza virus immunity in Southern China'). Written informed consent was obtained from all participants over 12 years old; verbal assent was obtained from participants 12 years old or younger. Written permission from a legally authorized representative was obtained for all participants under 18 years old.

### Cohort and serological data

We used serum collected from 777 participants who were recruited to an ongoing Fluscape cohort in Guangzhou, China, and provided blood samples for both a baseline visit (December 2009 to January 2011) and a follow-up visit (June 2014 to June 2015) (*Jiang et al., 2017*; *Yang et al., 2020*). The cohort recruited 40 locations that are randomly distributed in a fan-shaped area spanning from the city center to the neighboring rural areas. Participants, aged 2–86 years old with a male-to-female ratio of 1.106:1 at baseline sampling, were recruited from households that were randomly selected in these locations. Details of the cohort and participants included have been described previously (*Jiang et al., 2017*; *Yang et al., 2020*).

We measured antibody titers against 21 A(H3N2) strains using HI assays of paired serum collected from the two visits (*Yang et al., 2020*). Strains tested were isolated from 1968 to 2014, and priority was given to those included in vaccine formulation and/or used to construct the antibody landscape by *Fonville et al., 2014*. The strains we used are A/Hong Kong/1968, X-31 (isolated in 1970), A/England/1972, A/Victoria/1975, A/Texas/1977, A/Bangkok/1979, A/Philippines/1982, A/Mississippi/1985,A/Sichuan/1987, A/Beijing/1989, A/Beijing/1992, A/Wuhan/1995,A/Victoria/1998, A/Fujian/2000, A/Fujian/2002, A/California/2004, A/Brisbane/2007, A/Perth/2009, A/Victoria/2009, A/Texas/2012, and A/Hong Kong/2014 (*Yang et al., 2020*). These virus strains were obtained through the World Health Organization (WHO) collaboration network and passaged on Madin–Darby Canine Kidney (MDCK, ATCC CCL-34) cells or 9-day-old embryonic chicken eggs. Detailed laboratory methods have been described previously (*Lessler et al., 2011*; *Yang et al., 2020*).

## Statistical analysis

### Generalized additive model

To extract population- from individual-level A(H3N2) activity, we fitted a GAM of log HI titers (*Figure 1A*, *Figure 1—figure supplement 1A*) on the spline of age at baseline sampling, the spline of age at circulation (i.e., difference between year of strain isolation and year of birth of the participant) with strain-specific intercepts, which has been described in detail in a previous study (*Lessler et al., 2012*). In brief, log-titer for strain $j$ and participant $i$ is modeled as

$$E\left(logT_{i,j}\right) = \beta_{0,j} + \beta_1 s\left(a_i\right) + \beta_2 s\left(a_i - y_j\right) \tag{1}$$

where $s\left(.\right)$ denotes spline terms, $a_i$ denotes the age of the participant $i$ at baseline sampling, and $y_j$ denotes the number of years since strain $j$ was isolated until baseline sampling. Strain-specific intercepts $\beta_{0,j}$ were estimated and further used as a proxy for the population-level variations in A(H3N2) activities between 1968 and 2014 in the main analysis (*Figure 1B*, *Figure 1—figure supplement 1B*).

Residuals were calculated as the difference between observed and predicted log-titers from the fitted GAM to characterize individual-level A(H3N2) immune responses (*Figure 1C*, *Figure 1—figure supplement 6A*). A time series of residuals for participant $i$ was derived as

$$R_i\left(j\right) = logT_{i,j} - E\left(logT_{i,j}\right) \tag{2}$$

and then chronologically ordered by the year of strain isolations. HI titers for baseline and follow-up visits were fitted separately, and only titers to strains that were isolated after the person was born were included in the model.

### Fourier analysis

Periodicity in individual antibody responses to influenza was examined using Fourier spectral analysis with linear detrending, from which variances explained by each frequency were extracted (*Figure 1D*, *Figure 1—figure supplement 6B*). As the tested A(H3N2) strains were irregularly spaced in time (2–3-year intervals), we fitted a spline and interpolated the time series to a yearly resolution before applying the Fourier analysis.

For individual-level periodicity, we extracted the frequency that explained the most variance (i.e., the greatest spectral power; 'peak frequency' hereafter) for each individual (*Figure 1D*, *Figure 1—figure supplement 6B*) and plotted the distribution of peak frequencies across 777 participants (*Figure 1E*, *Figure 1—figure supplement 1C*). To test the significance against the null distributions, we compared the observed distribution of peak frequencies with the distribution of peak frequencies from 1000 permutations. In each permutation, we shuffled the time series of residuals for each person and extracted the peak frequency for each individual (*Figure 1—figure supplement 6C and D*).

## Validations and sensitivity analyses

### Weighted frequency

The peak frequency of the Fourier spectrum we extracted only represents the frequency that explained the most variance, but it cannot reflect the variance explained by the other frequencies, that is, whether the Fourier spectrum is skewed toward the peak frequency or is flatly distributed (*Figure 1D*, *Figure 1—figure supplement 6*). Thus, we calculated the average frequency weighted by the variance explained ('weighted frequency'), to represent the weighted center for each spectrum (*Figure 1—figure supplement 6B*). The weighted frequency $f_w$ was calculated as

$$f_w = \frac{\sum_k f_k v_k}{\sum_k v_k} \tag{3}$$

where $f_k$ and $v_k$ denote the $k^{th}$ examined frequency and its estimated variance the Fourier spectrum. We found that our data was more likely to show lower weighted frequencies and longer periods compared to the permutations (*Figure 2A*).

## Addressing irregularly sampled intervals with Lomb–Scargle periodogram

To examine the impact of using irregularly sampled intervals and interpolation on the results from the Fourier spectrum analysis, we performed a sensitivity analysis using the Lomb–Scargle periodogram (*Glynn et al., 2006*), which is often used to detect the periodicity of irregularly sampled time series. As in the previously described Fourier spectrum analysis, we derived the spectrum for each individual's time series of residuals using the Lomb–Scargle periodogram, which estimated the variance explained at each frequency. We then extracted the frequency with the most variance explained, that is, 'peak frequency' for each spectrum. Similar to the main analysis, we compared the distribution of observed peak frequency derived from Lomb–Scargle periodogram across participants with those from 1000 permutations (*Figure 2B*). In each permutation, we shuffled the time series of residuals for each individual (maintaining the irregularity in the sampling), and then extracted the peak frequency of the Lomb–Scargle periodogram for each shuffled time series.

## Removing nonlinear trends with empirical mode decomposition (EMD)

Although we removed the linear trend before applying Fourier analysis, several time series contained nonlinear trends that could potentially bias the estimate of the peak frequency to lower values (e.g., participants 1 and 2 in *Figure 2—figure supplement 1*). In order to avoid this issue, we performed Fourier analysis with the time series of residuals after removing nonlinear trends using EMD (*Huang et al., 1998*).

To do this, we first applied EMD to each individual's time series of residuals, and extracted the underlying trend, defined as the 'residue' remaining after all intrinsic mode functions have been extracted (*Figure 2—figure supplement 1*). We then detrended the time series by subtracting this 'residue' from the original time series. Finally, the peak frequency of the Fourier spectrum of the detrended time series was extracted for the individual, and the distribution of peak frequencies was plotted across individuals (*Figure 2C*). For each permutation, we shuffled the individual's time series and applied EMD to the shuffled time series. The remaining steps for the permutation analysis were the same as above.

For participants whose time series showed nonlinear trends (e.g., participants 1 and 2 in *Figure 2—figure supplement 1*), peak frequency shifted to a higher frequency after detrending with EMD. Meanwhile, for participants whose time series showed cycles (e.g., participants 3 and 4), the low-frequency cycles were no longer detectable after detrending with EMD. Therefore, the results shown in *Figure 2B* were the distribution of peak frequencies after removing both nonlinear trends and some low-frequency cycles. The 20–40-year cycles were still detectable for both visits, suggesting that the long-term cycle we detected was not solely explained by the nonlinear trend of the time series.

## Dropping every other strain

In order to test whether the reported cycles in the individual residuals were influenced by the relatively stronger responses to some strain (e.g., X-31, A/Mississippi/1985, A/Beijing/1992, and A/Fujian/2002), we dropped 1 out of the 21 strains and repeated the Fourier analysis to the time series of the remaining 20 strains. For the permutation test, we shuffled the time series of the remaining 20 strains and reinterpolated the shuffled time series for each individual. Results suggested that dropping out one strain did not affect our conclusions (*Figure 2—figure supplements 2 and 3*).

## Validations using random values or values from periodic curves

We tested the robustness of our results from the Fourier analysis with a time series of 21 irregularly sampled data points with the same time resolution as our data. Time series consisted of random values generated from varying underlying distributions. Briefly, we drew a set of random values for each individual and the length of time series was based on the individual's year of birth. We performed the interpolations, Fourier analysis, and extracted the peak frequency of the Fourier spectrum for each new time series. Finally, the distribution of peak frequencies for the simulated time series and their null distributions from permutations were compared.

We performed this analysis using values drawn from normal and lognormal distributions without periodicity (*Figure 1—figure supplement 7A and B*). In addition, we randomly replaced 2–4 points in each individual's time series with outlier values that are rare in the underlying distribution in order

to mimic the relatively higher titers to several strains observed in the data (*Figure 1—figure supplement 7C and D*). There were no significant differences between peak distributions of the simulated random time series and their permutations, suggesting that the low frequencies identified in our real data cannot be explained by the correlation structure introduced by irregularly sampled intervals, interpolation, randomness, and outlier values.

We then applied the Fourier analysis on time series generated by sampling from periodic curves with white noise. To do this, we first simulated a time series from 1968 to 2014 on a yearly basis for each participant from a sinusoidal curve with a certain periodicity and white noise. We then subset the simulated time series to the years when our tested A(H3N2) strains were isolated relative to each participant's year of birth. We applied the previously described interpolation and Fourier analysis to the subset of each time series. We repeated the above analysis for 777 participants and compared the distributions of peak frequencies from simulated time series and their 1000 permutations (*Figure 1—figure supplement 8*). Four scenarios were tested: (1) time series of all participants had a single 25-year periodicity; (2) time series of all participants had a single 16-year periodicity; (3) time series of half of the participants had a single 25-year periodicity, and time series of the other half of participants had a single 16-year periodicity; and (4) time series of all participants contained two superimposed periodic curves, with periodicities of 25 and 5 years. Results suggested that the method we used in the main analysis can uncover the real low-frequency signals, while uncovering high-frequency signals could be challenging due to the resolution of our data.

## Excluding participants who were born after 1968

To examine the effects of participants who had a relatively shorter exposure history of A(H3N2) on the reported cycles, we repeated the Fourier spectrum analysis with time series of residuals for a subset of participants (n = 487) who could have experienced all tested A(H3N2) strains, that is, born before 1968. The analysis follows the same steps as the main analysis except that the distributions of peak frequencies were plotted across 487 eligible participants. Cycles with low frequencies were found for the subset of senior participants as well, with an increasing proportion of participants having the lowest frequency (*Figure 2—figure supplement 4*).

## Sera from Vietnam study

In order to test our results with a different population, we repeated the analysis with publicly available data reported in a previous Vietnam study (*Bedford et al., 2014*; *Kucharski et al., 2015*). Longitudinal sera were collected for 69 participants in Ha Nam, Vietnam. Participants were aged 7–95 years in 2012, of which 48% were under 30s (*Fonville et al., 2014*). Sera were repeatedly collected from these participants between 2007 and 2012 on a yearly basis (*Fonville et al., 2014*). HI titers were measured for 57 A(H3N2) strains isolated between 1968 and 2011, with a finer resolution in the more recent years (*Fonville et al., 2014*).

In the Vietnam study, multiple strains had been isolated in the same year, resulting in multiple titers being available for a given year for each individual. Therefore, we fitted a cubic spline in order to derive a time series that captured the geometric mean titers to strains isolated in the same year. We then applied Fourier analysis to each splined time series and extracted the peak frequency of each spectrum. The distribution of peak frequencies was characterized across 69 individuals by the year of serum collection (*Figure 2—figure supplement 5*). For the permutation analysis, HI titers were shuffled before fitting splines to the time series. As the age of participants was not available, we performed the analysis with raw titers without adjustment on age. Significant cycles with frequencies ranging from 0.050 to 0.075 (~13–20 years) were detected for serums collected in 2007, 2009, 2010, and 2011, coinciding with the frequencies detected using the raw titers of serums collected in our baseline visit.

## Simulations of life-course infection history and immune responses
### Model descriptions

In order to explore the mechanisms behind the reported dynamics of human immune responses to influenza, we applied a previously described mechanistic model (*Kucharski et al., 2018*) to generate realizations of lifelong infection history and subsequent immune responses. Simulations were individually based on a yearly scale and returned as antibody profiles consisting of titers to a panel of 47

strains (i.e., strains isolated from 1968 to 2014) that were tested in 2014. The simulations consisted of the following steps:

1. *Construct initial antibody profile.* An initial antibody profile was generated for the sera collected in 1968 for participants who were born on or before 1968, or the year of birth for participants who were born after 1968. Titers to all 47 strains were assumed to be 0 for initial antibody profiles.
2. *Extract preexisting titers for each season.* For an examined year $y$, we extracted the titer to the strain that was isolated in $y$ from the latest antibody profile (i.e., antibody profile measured in year $y$) (*Figure 3—figure supplement 1*).
3. *Determine the probability of infection of the circulating strain.* The probability of an individual infected by the strain isolated in year $y$ was calculated according to the immunity-dependent protection (*Equation 11*; see section 'Modeling immunity-dependent protection' for details) and annual A(H3N2) activity (*Figure 3—source data 1*). For the initial year (i.e., 1968), we imposed a pandemic with an attack rate of 50% in the main analysis. The strain isolated in year $y$ was assumed to be the circulating strain of that year.
4. *Simulate infection event.* Infection outcome was randomly generated following a binomial distribution with the probability calculated in step 3. Infection outcomes were simulated for each individual every year.
5. *Update immune responses.* Immune responses (i.e., boost and cross-reactions from infections and/or immunity decay) to the whole panel of strains were updated based on the annual infection outcome (*Figure 3—figure supplement 1*) using the previously described model and estimates (*Equations 4–9* and *Figure 3—source data 1*; see section 'Modeling immune responses' for details; *Kucharski et al., 2018*). The updated antibody profiles are then used in step 2 for the following year (*Figure 3—figure supplement 1*).
6. Repeat steps 2–5 until 2014 and extract the antibody profiles measured in 2014.

We simulated antibody profiles for 777 individuals of the same ages as the participants in our study. For each individual, we repeated the above six steps from 1968 (or the year of birth) to 2014 and extracted the antibody profiles to all 47 strains in 2014 for further analyses.

In order to explore the mechanisms that created the observed cycles, we performed simulations under different scenarios that considered several generally recognized components of immunity (*Figure 3I*):

1. Baseline scenario (*Figure 3A*), which assumed a constant 50% annual probability of infection for all individuals and no cross-reaction or cross-protection from past infections.
2. Population activity-only scenario (*Figure 3B*), which assumed a random varied population-level viral activity that would affect individual probability of infection, and no cross-reaction or cross-protection from past infections.
3. Narrow cross-reaction scenario (*Figure 3C*), which assumed annual individual probability of infection would be determined by individual preexisting titer and a random varied population-level viral activity, and cross-reactions only to a narrow range of antigenic relatives (i.e., recent strains).
4. Broad cross-reaction scenario (*Figure 3D*), which assumed annual individual probability of infection would be determined by individual preexisting titer and a random varied population-level viral activity, and cross-reactions only to a broad range of antigenic relatives (i.e., distant strains).
5. Cross-reaction-only scenario (*Figure 3E*), which assumed a constant 50% annual probability of infection for all individuals, and cross-reaction to both narrow and broad range of antigenic relatives, but no cross-protection from past infections.
6. No population activity-only scenario (*Figure 3F*), which assumed annual individual probability of infection would be determined by individual preexisting titer but not population-level viral activity, and cross-reactions and cross-protection to both narrow and broad range of antigenic relatives.
7. Random population activity scenario (*Figure 3F*), which assumed annual individual probability of infection would be determined by individual preexisting titer and a random varied population-level viral activity, and cross-reactions and cross-protection to both narrow and broad range of antigenic relatives.
8. Periodic population activity scenario (*Figure 3F*), which assumed annual individual probability of infection would be determined by individual preexisting titer and a periodically varied population-level viral activity (5-year periodicity), and cross-reactions and cross-protection to both narrow and broad range of antigenic relatives.

## Modeling immune responses

We adapted the previously described model to simulate immune responses after exposures (**Kucharski et al., 2018**); the parameters used are shown in **Figure 3—source data 1**. The immune response after an infection is divided into long-term boosting, $d_l(j, m_t)$, and short-term boosting, $d_s(j, m_t)$, modeled as

$$d_l(j, m_t) = max(0, 1 - \sigma_l \delta_{j, m_t}) \tag{4}$$

$$d_s(j, m_t) = max(0, 1 - \sigma_s \delta_{j, m_t}) \tag{5}$$

where $\delta_{j, m_t}$ denotes the difference in antigenic difference between strain $j$ and the previously infecting strain $m_t$:

$$\delta_{j, mt} = \rho T_{j, m_t} \tag{6}$$

$T_{j, m_t}$ denotes the number of years between when the tested strain $j$ and the infected strain $m_t$ were isolated, and $\rho$ is the rate of change in antigenic units per year. Parameters $\sigma_l$ and $\sigma_s$ represent the durations of cross-reactions. Short-term immunity also wanes, as set by the waning duration $\omega$ and the number of years between the year of infection by strain $m_t$ and year of testing ($T_{m_t}$):

$$w(m_t) = max(0, 1 - \omega T_{m_t}) \tag{7}$$

The antigenic seniority was scaled by a suppression parameter $\tau$ and the order of infection ($N_m$) among all infected strains $X_t$:

$$s(X_t, m_t) = max(0, 1 - \tau(N_m - 1)) \tag{8}$$

Prior study estimated $\tau$ as 0.04, while we explored both 0 and 0.04 and found minimal impact on our main results. Therefore, we assumed $\tau$ as 0 for simplicity.

Finally, the titer against strain $j$ for person $i$ tested in year $t$ is

$$\mu_{i,j,t} = \sum_{m_t \in X_t} s(X_t, m_t) \left[ \mu_l d_l(j, m_t) + \mu_s w(m_t) d_s(j, m_t) \right] \tag{9}$$

where $\mu_l$ and $\mu_s$ denote the mean log-titers of long-term and short-term boost to an infecting strain, respectively.

## Modeling immunity-dependent protection

For the baseline scenario and cross-reaction-only scenario, the probability of infection was assumed to be a constant. For the cross-protection and antigenic seniority scenarios, a higher HI titer to a circulating A(H3N2) strain is assumed to be associated with lower risk of infection with that strain (**Figure 3—figure supplement 1**). We assumed that the 50% protective titer is 1:40 (i.e., $\mu_{50} = 3$ on a log scale). The titer-dependent risk of infection is modeled as (**Vieira et al., 2021**)

$$p_{I|\mu} = \frac{1}{1 + e^{\beta(\mu - \mu_{50})}} \tag{10}$$

where $\beta$ is the scale parameter of the titer-dependent protection estimated in previous studies (**Yuan et al., 2017**). After adjusting for annual A(H3N2) activity ($\lambda_t$), the titer-dependent probability of infection of strain $j$ for person $i$ tested in year $t$ s

$$p_{I|\mu_{i, j, t}} = \frac{\lambda_t}{1 + e^{\beta(\mu_{i, j, t} - \mu_{50})}} \tag{11}$$

The annual A(H3N2) activity, $\lambda_t$, was included to explore the impact of the virus circulation at populethical level on the observed long-term cycles in individual antibody responses. Three different hypothetical scenarios were assumed for $\lambda_t$:

1. $\lambda_t = 0.2$, where annual activity was assumed as constant across the 47 years with an annual attack rate of 20%.
2. $\lambda_t \sim Uniform(0, 0.2)$, where annual activity varies between 0 and 0.2 randomly.
3. $\lambda_t = 0.2sin(2\pi t/5)$, where annual activity varies year to year with a periodic pattern.

The main objective of this analysis was to demonstrate that population circulation alone was not able to recover the observed periodicity in individual antibody responses. Thus, although there remain debates about the interactions between influenza subtypes, we showed that it seemed not to be the main driver of the observed periodicity in individual antibody responses.

## Prediction of individual antibody responses to future strains using intrinsic cycles

### Estimation of the phase

We estimated the phase angle ($p_{y,i}$, in degree) of antibodies against a strain $j$ that circulated in a given year $y$ for person $i$, to represent the position where the antibody against the tested strain stands in the entire antibody responses of this person. We first fitted a regression to the time series of individual residuals ($R_i(y)$) for strains that were isolated during a certain period and included harmonic terms that represent the periodic patterns in the antibody responses.

$$R_i(y) = \gamma_0 + \gamma_1 sin(2\pi yf) + \gamma_1 cos(2\pi yf) \tag{12}$$

where $f$ is assumed as the inverse of the periodicity that most of our participants showed, that is, 24 years. With the estimated coefficients from *Equation 12*, we predict the phase angle in radian ($r_{y,i}$) of strain $j$ that circulated in given year $y$ for person $i$ as follows:

$$r_{y,i} = 2\pi yf_0 + \phi_i \tag{13}$$

where $\phi_i$ denotes the person $i$ 's phase shift:

$$\phi_i = atan2(\gamma_1, \gamma_2) \tag{14}$$

We then translated the phase angle from radian to degree as follows:

$$p_{y,i} = \frac{180° \times r_{y,i}}{\pi} \tag{15}$$

The phase angle in degree was then classified into four categories, namely, phase I (0–90°), II (91–180°), III (181–270°), and IV (271–360°) (*Figure 4A*). Of note, we fitted the model aiming to estimate the position of the harmonic oscillators and did not consider for other nonharmonic factors; therefore, the model may not fully capture the variations of the data.

### Comparison between observed and predicted phase in 2012

We predicted the phase angle (in degree) for the strain that circulated in 2012, which is the middle between our baseline (2010) and follow-up (2014) visit. We first fitted *Equation 12* to HI titers that were measured for strains isolated between 1968 and 2002 measured at baseline (i.e., 14 strains, *Figure 4B*). Predicted phase in 2012 was then estimated using *Equations 13–15*. To estimate the observed phase in 2012, we fitted the model in *Equation 12* to the full panel of tested strains (i.e., 21 strains) measured at baseline and calculated the phase angle using *Equations 13–15*.

To assess the consistency between the prediction and observation of phases in 2012, we plotted the distribution of the observed phase in 2012 among people who were predicted to in each of the four phases in 2012 (*Figure 4C*).

### Association between phase and seroconversion

We examined the association between the phase in individual antibody responses and antibody responses to circulating strains (*Figure 4D*). We measured the antibody responses circulating strains as the seroconversion (i.e., fold of change ≥4) to either A/Texas/2012 or A/HongKong/2014 (i.e., strains that were circulated between baseline and follow-up). We fitted a logistic regression to seroconversion and adjusted for the predicted phase (in categories) in 2012, the average of titers against the two tested strains at baseline (i.e., preexisting titers in log scale), and the participants' age at baseline.

## Disentangle birth cohort effects using intrinsic cycles in individual antibody responses

To examine the differences in phase distribution across different birth cohorts, we first estimated the observed individual phase in 2012 by fitting *Equations 12–15* to the full panel of tested strains measured at baseline. We compared the distribution of phase in 2012 among 5-year binned birth cohorts using chi-squared test (*Figure 4E*).

We estimated the phase distribution across birth cohorts using the predicted phase in 2012, which was derived by fitting *Equations 12–15* to the HI titers against strains isolated between 1968 and 2002 measured at baseline. We examined the association between the predicted and observed phase distribution across birth cohorts by calculating the Pearson correlation between the predicted and observed proportion of phase IV in each birth cohort (*Figure 4F*).

We examined the association between the predicted cohort-specific proportion of phase IV and the observed proportion of seroconversion to either A/Texas/2012 or A/HongKong/2014 using Pearson correlation (*Figure 4G*).

## Software and programs

The studies were performed following the STROBE checklist wherever is applicable. All analyses were performed in R version 4.1.0 (R Foundation for Statistical Computing, Vienna, Austria). We used the 'mgcv' package to fit GAMs (*Wood, 2011*). The Lomb–Scargle periodogram was performed with the 'spectral' package (*Seilmayer, 2016*). We performed the empirical mode decomposition with the 'EMD' package (*Kim and Oh, 2009*). Simulations of life-course infection history and immune responses were performed with the 'Rcpp' package (*Eddelbuettel and François, 2011*). Source code used in this study is openly available at https://github.com/UF-IDD/Fluscape_Periodicity, (copy archived at swh:1:rev:2fa04290f2749633f1b236be3f7fd36b33aee954; *Yang, 2022*).

## Acknowledgements

This study was supported by grants from the NIH R56AG048075 (DATC, JL), NIH R01AI114703 (DATC, BY), and the Wellcome Trust 200861/Z/16/Z (SR) and 200187/Z/15/Z (SR). This work was also supported by research grants from Guangdong Government HZQB-KCZYZ-2021014 and 2019B121205009 (YG and HZ). DATC, JMR, and SR acknowledge support from the National Institutes of Health Fogarty Institute (R01TW0008246). JMR acknowledges support from the Medical Research Council (MR/S004793/1) and the Engineering and Physical Sciences Research Council (EP/N014499/1).

## Additional information

### Competing interests

Bernardo García-Carreras: BGC received financial research support through his institution from Merck for unrelated work. Justin Lessler: JL receives research support from CDC and NIH-NIGMS for for unrelated work. Steven Riley: SR receives grants from Wellcome Trust. Derek A Cummings: DATC received financial research support through his institution from Merck for unrelated work. The other authors declare that no competing interests exist.

### Funding

| Funder | Grant reference number | Author |
| --- | --- | --- |
| National Institute on Aging | R56AG048075 | Justin Lessler |
| Wellcome Trust | 200861/Z/16/Z | Steven Riley |
| National Institute of Allergy and Infectious Diseases | R01AI114703 | Derek A Cummings |
| Guangzhou Government | 2019B121205009 | Huachen Zhu Yi Guan |

| Funder | Grant reference number | Author |
|---|---|---|
| Guangzhou Government | HZQB-KCZYZ-2021014 | Huachen Zhu<br>Yi Guan |
| National Institutes of Health | R01TW0008246 | Jonathan M Read |
| Wellcome Trust | 200187/Z/15/Z | Steven Riley |
| Medical Research Council | MR/S004793/1 | Jonathan M Read |
| Physical Sciences Research Council | EP/N014499/1 | Jonathan M Read |

The funders had no role in study design, data collection and interpretation, or the decision to submit the work for publication. For the purpose of Open Access, the authors have applied a CC BY public copyright license to any Author Accepted Manuscript version arising from this submission.

## Author contributions

Bingyi Yang, Conceptualization, Data curation, Software, Formal analysis, Validation, Investigation, Visualization, Methodology, Writing – original draft, Writing – review and editing; Bernardo García-Carreras, C Jessica E Metcalf, Methodology, Writing – review and editing; Justin Lessler, Conceptualization, Resources, Data curation, Supervision, Funding acquisition, Methodology, Writing – review and editing; Jonathan M Read, Conceptualization, Resources, Supervision, Methodology, Writing – review and editing; Huachen Zhu, Resources, Validation, Investigation, Writing – review and editing; James A Hay, Software, Methodology, Writing – review and editing; Kin O Kwok, Resources, Investigation, Writing – review and editing; Ruiyun Shen, Investigation, Writing – review and editing; Chao Q Jiang, Conceptualization, Supervision, Project administration, Writing – review and editing; Yi Guan, Resources, Supervision, Writing – review and editing; Steven Riley, Conceptualization, Resources, Data curation, Supervision, Funding acquisition, Methodology, Project administration, Writing – review and editing; Derek A Cummings, Conceptualization, Resources, Data curation, Formal analysis, Supervision, Funding acquisition, Validation, Investigation, Visualization, Methodology, Writing – original draft, Project administration, Writing – review and editing

## Author ORCIDs

Bingyi Yang ![ORCID] http://orcid.org/0000-0002-0811-8332
C Jessica E Metcalf ![ORCID] http://orcid.org/0000-0003-3166-7521
James A Hay ![ORCID] http://orcid.org/0000-0002-1998-1844
Steven Riley ![ORCID] http://orcid.org/0000-0001-7904-4804

## Ethics

Human subjects: The following institutional review boards approved the study protocols: Johns Hopkins Bloomberg School of Public Health(IRB 1716), University of Florida (IRB201601953), University of Liverpool, University of Hong Kong (UW 09-020) and Guangzhou No. 12 Hospital ("Research on human influenza virus immunity in Southern China"). Written informed consent was obtained from all participants over 12 years old; verbal assent was obtained from participants 12 years old or younger. Written permission of a legally authorized representative was obtained for all participants under 18 years old.

## Decision letter and Author response

Decision letter https://doi.org/10.7554/eLife.81457.sa1
Author response https://doi.org/10.7554/eLife.81457.sa2

# Additional files

## Supplementary files

- MDAR checklist

## Data availability

All data generated or analysed during this study are included in the manuscript and supporting file; Source Data files have been provided for Figures 1 to 4.

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
