## [Editor Report]

This article follows the still unanswered concept of ‘original antigenic sin’ and shows the existence of a 24-year periodicity of the immune response against influenza H3N2. The valuable work suggests a long-term periodicity of individual antibody response to influenza A (H3N2) within a city.

---

## [Decision Letter]

**Decision letter after peer review:**

Thank you for submitting your article "Long term intrinsic cycling in human life course antibody responses to influenza A(H3N2)" for consideration by *eLife*. Your article has been reviewed by 2 peer reviewers, and the evaluation has been overseen by a Reviewing Editor and Betty Diamond as the Senior Editor. The following individuals involved in the review of your submission have agreed to reveal their identity: Koji Tokoyoda (Reviewer #1); Marcus Robinson (Reviewer #2).

Essential revisions:

1) The model needs to be tested with different parameters to show that a 24-year periodicity is best. In other words, what would happen to the predictions if the periodiciyty were 35-year or 6-year?

2) The strains inducing high HI titers may have similar mutations and may be reactive to the same antibodies. What are the mutation frequencies among 21 A(H3N2) strains? So, generating a map of antigenic distance for the strain used is required.

3) As seen in the 2) comment of reviewer #1, reinfection/vaccination at any time point confounds the overall titers and seroconversion. Hence, considering this point, authors need to make a more explicit discussion on how these are controlled.

*Reviewer #2 (Recommendations for the authors):*

This manuscript should be evaluated by a biomathematician to cover the appropriateness and form of the spline equations and the Fourier transform approach.

The introduction should discuss original antigenic sin (OAS) and its foundations and would benefit from discussing anamnestic Ig responses against previously encountered strains a little more than it does currently.

L95: (Please insert 'and' between '2021),' and 'antigenic'.

---

## [Author Response]

Essential revisions:1) The model needs to be tested with different parameters to show that a 24-year periodicity is best. In other words, what would happen to the predictions if the periodiciyty were 35-year or 6-year?

Thank you for your comments. As suggested we assumed a 35-year and 6-year periodicity and repeated analyses in the “Disentangling cohort effects using cycles in individual antibody responses” section (Figure 4F-G). Results suggested that model predictions with either 35-year or 6-year periodicity failed to outcompete the model predictions assuming a 24 year periodicity (Figure 4—figure supplement 1). For example, the observed proportion of seroconversion to circulating strains in each cohort have correlation coefficients of 0.49 (p-value = 0.05), 0.63 (p-value = 0.02) and -0.12 (p-value = 0.69) with the predicted proportion of phase IV when assuming a 35-, 24- and 6-year periodicity, respectively. The higher correlation and statistical significance suggests that the 24 year periodicity has greater association with the proportion who seroconvert.

We also hope to clarify that the 24-40 year periodicity (i.e., frequency ranging from 0.025-0.05) was identified from the observed data using Fourier spectrum analyses, which is robust to a number of analytic and sampling variations. The 24-year periodicity was used as it was shared by most of the participants (Figure 1 – table supplement 1). While we acknowledge that the exact value for this long-term periodicity may depend on the number and span of the tested strains, we had strong support for this cyclic variability in our data. For analyses in Figure 4, we also used an individual periodicity that was determined by each individual’s peak frequency (i.e., not a unified 24-year periodicity for all) and found consistent results, further supporting that the 24-year periodicity was shared by most of our participants. We included this discussion in the main text (lines 337-340).

To address this comment, we also included the results of analyses using alternative 35- and 6-year periodicity (similar to Figure 4 which shows the 24 year periodic results) in Figure 4 —figure supplement 1, and reported these results in the main text (lines 262-264).

2) The strains inducing high HI titers may have similar mutations and may be reactive to the same antibodies. What are the mutation frequencies among 21 A(H3N2) strains? So, generating a map of antigenic distance for the strain used is required.

Thank you for your comments. Our selection of the 21 tested strains were selected to span the circulation duration of A(H3N2) strains since 1968, when this subtype emerged. We also prioritized strains that were included in vaccine formulations as well as those tested to create the antigenic map by Fonville et al. [1].

As suggested by the editors and reviewers, we reproduced the antigenic map (up to strains isolated in 2010) using previously published data [1]. We compared this with our tested A(H3N2) strains in Figure 1—figure supplement 3. The figure shows that the 21 strains (or their belonging antigenic clusters if the strains were not used for the map) largely tracked the antigenic evolution of A(H3N2) since its emergence in 1968, with a reported mutation rate of 0.778-unit changes in antigenic space per year [1, 2].

We further calculated the paired antigenic distance of strains tested in the antigenic map, which was highly correlated with the time interval between the isolation of the two strains. The figure also suggested our tested strains cover the time spans and antigenic distances that were shown in the original antigenic map. In addition, our observed periodicity was identified in individual time series of residuals, which has removed the shared virus responses or assay measurement effects (Figure 1). Therefore, we believe that the impact of specific mutations may have been limited in our findings.

To address this comment, we included the reproduced antigenic map showing the locations of the tested strains and their pair-wise antigenic distance in Figure 1—figure supplement 3 and referenced in the main text (lines 127).

3) As seen in the 2) comment of reviewer #1, reinfection/vaccination at any time point confounds the overall titers and seroconversion. Hence, considering this point, authors need to make a more explicit discussion on how these are controlled.

Thank you for your comments. We agree with the editor and reviewers that the observed seroconversion of the circulating strains may reflect responses after vaccination or recent infections. The overall influenza vaccination coverage in China has been <5% [3, 4]. Only 1.3% (10 out of 777) of our participants self-reported with any influenza vaccination history at baseline, and only 5 individuals reported vaccination at follow-up [5]. Therefore, we believe that the observed periodicity and seroconversion pattern in our study was unlikely to be attributed to recent influenza vaccinations.

Instead, we believe that the observed seroconversions to circulating strains between our baseline and follow-up were likely due to pervasive exposure or infection with influenza (here, we admit the possibility that exposure without productive infection could provide a stimulus to the immune response); 98% and 74% of participants experienced 2- and 4-fold rise to any of the 21 tested A(H3N2) strains [5]. However, recent re-exposures or re-infections would not substantially affect our observations on the long-term periodicity in individuals antibody responses. The strongest evidence for this is that we found similar periodic behaviour in the two time points of sample collection that we report in this manuscript. We repeated the Fourier analyses for serum collected at follow-up (i.e., may have been experienced re-exposures or reinfections), and found that the long-term periodicity was still detectable (Figure 1—figure supplement 1C).

In fact, we believe that repeated exposure contributes to the periodic phenomenon that we report. Repeat exposures to A(H3N2) have been found to subsequently lead to antibody responses that were affected by pre-existing antibodies and cross-reaction [1, 6]. We incorporate re-infections in our simulations, with and without subsequent cross-reaction with previously exposed distant strains (Figure 3I). Results indicate that reinfection alone cannot recover the observed long-term periodicity (Figure 3A), while reinfection plus the resulting cross-reaction can recover such long-term periodicity (Figure 3D).

In summary, we believe that repeated exposures or re-infections are integral to the generation of our reported periodicity, hypothesizing that they may be drivers of life-course antibody profiles and the observed periodicity. However, exposures proximal to serum collection do not appear to substantially affect the observed periodicity, as we see consistent results from sera taken from individuals ~3-4 years apart.

To address this comment, we reported the vaccination status of our participants when introducing the data (lines 127-129) and in the discussions (lines 280-282 and 313-315):

“Only 0.6% (n = 5) of participants self-reported influenza vaccinations between the two visits, therefore, the observed changes in HI titers between the two visits were likely due to natural exposures.”

“Due to the low influenza vaccine coverage in our participants and in China in general, the observed seroconversions likely reflected antibody responses after natural exposures during the study period.”

“Particularly, our simulation results suggested that model included repeated exposures or population level A(H3N2) activity alone did not recover the long-term periodicity (Figure 3).”

Reviewer #2 (Recommendations for the authors):This manuscript should be evaluated by a biomathematician to cover the appropriateness and form of the spline equations and the Fourier transform approach.

Thank you for your comments. The core model and parameter values that we used in the study has been previously published in the peer-reviewed journals. The Fourier transform is a classic method with several widely acknowledged R packages available. To insure the appropriateness of the options of the Fourier transform approach, we conducted several sensitivity analysis to check the robustness of our results. Specifically, we obtained similar results when using different spline methods and a different transform approach (Figure 2B). Simulation-and-recovery experiments suggested that power of our Fourier transform approach and suggesting that the observed periodicity was not able to be recovered in non-periodic time series (i.e. a permuted version of the data) (Figure 1—figure supplement 7-8).

The introduction should discuss original antigenic sin (OAS) and its foundations and would benefit from discussing anamnestic Ig responses against previously encountered strains a little more than it does currently.

Thank you for your comments. We expanded our introduction with more discussion about OAS in the second paragraph.

“Original antigenic sin (OAS) is a widely accepted concept describing the hierarchical and persistent memory of antibodies from the primary exposure in childhood. Recent studies suggested that non-neutralizing antibodies acquired from previous exposures can be boosted and may blunt the immune responses to new infections.”

L95: (Please insert 'and' between '2021),' and 'antigenic'.

Revised, thank you.